# Pharmacological Evaluation of Novel Hydrazide and Hydrazone Derivatives: Anti-Inflammatory and Analgesic Potential in Preclinical Models

**DOI:** 10.3390/molecules30071472

**Published:** 2025-03-26

**Authors:** Hristina Zlatanova-Tenisheva, Stanislava Vladimirova

**Affiliations:** 1Department of Pharmacology and Clinical Pharmacology, Medical University of Plovdiv, 4002 Plovdiv, Bulgaria; 2Department of Organic Synthesis, University of Chemical Technology and Metallurgy, 1756 Sofia, Bulgaria; vladimirova.s@uctm.edu

**Keywords:** hydrazones, tail-flick, paw withdrawal, formalin model, paw edema

## Abstract

Hydrazones, characterized by their C=N–NH functional group, are promising candidates in medicinal chemistry due to their ability to interact with biological targets. This study evaluated the anti-inflammatory and analgesic properties of *N*-pyrrolylcarbohydrazide (**1**) and four pyrrole hydrazone derivatives (**1A–D**) in male Wistar rats (6 weeks old). Anti-inflammatory activity was assessed using a carrageenan-induced paw edema model, while formalin, tail flick, and paw withdrawal tests evaluated analgesia. Compound **1** exhibited dose-dependent anti-inflammatory activity. At 20 mg/kg, significant edema reductions were observed at the 2nd (*p* = 0.035) and 3rd hours (*p* = 0.022), while at 40 mg/kg, reductions remained significant at the 2nd (*p* = 0.008) and 3rd hours (*p* = 0.046). Compound **1A** showed pronounced effects at 20 mg/kg at the 2nd (*p* = 0.005), 3rd (*p* < 0.001), and 4th hours (*p* = 0.004). Other compounds demonstrated minimal or no activity. Analgesic evaluation revealed that at 40 mg/kg, compound **1** significantly reduced paw-licking time in the second phase (*p* = 0.038). Compounds **1B**, **1C**, and **1D** exhibited transient effects in the first phase only (*p* < 0.05). Compound **1A** lacked significant analgesic activity. The findings suggest that structural modifications may enhance efficacy for broader therapeutic applications.

## 1. Introduction

Hydrazones, characterized by their unique C=N–NH functional group, are a versatile class of organic compounds widely explored in medicinal chemistry due to their structural flexibility and broad spectrum of biological activities. The hydrazone moiety is capable of hydrogen bonding and coordination with metal ions, making these compounds very useful for interactions with biological targets in drug design [1]. Hydrazone derivatives exhibit a wide range of biological and pharmacological properties. These include antimicrobial, anti-inflammatory, analgesic, antifungal, anticancer, antimalarial, anticonvulsant, and cardio-protective effects, among others [2,3,4,5]. Their incorporation into approved pharmaceutical products underlines their importance and potential for further development across a wide range of therapeutic areas.

Inflammation and pain are among the most common and debilitating symptoms associated with numerous pathological conditions, ranging from acute injuries to chronic diseases such as arthritis and neuropathies. These processes are complex and involve the interplay of pro-inflammatory cytokines, enzymes, and mediators such as prostaglandins, histamine, and bradykinin. Current pharmacological interventions, including non-steroidal anti-inflammatory drugs (NSAIDs) and opioids, target these pathways effectively but are often associated with serious side effects. For example, NSAIDs are known for causing gastrointestinal irritation, renal toxicity, and cardiovascular risks, while opioids are linked to the development of tolerance, dependence, and abuse. These limitations outline the urgent need for the development of new therapeutic agents with better efficacy and safety profiles [6].

In this regard, hydrazones have appeared as a very promising class of compounds due to their ability to modulate key biochemical pathways involved in inflammation and pain. Their structural diversity allows for the fine tuning of the pharmacokinetic and pharmacodynamic properties to develop derivatives with targeted activities and fewer adverse effects [7,8]. Preliminary studies have demonstrated the potential of hydrazone-based compounds as anti-inflammatory and analgesic agents, but further research is needed to fully elucidate their mechanisms of action and therapeutic efficacy.

In this study, a series of newly synthesized pyrrole derivatives—*N*-pyrrolylcarbohydrazide (**1**) and pyrrole hydrazones (**1A**–**D**)—was evaluated for its anti-inflammatory and analgesic activities using well-established animal models. The carrageenan-induced paw edema test was employed to investigate its efficacy in reducing inflammation, while the formalin, tail flick, and paw withdrawal tests were used to assess both central and peripheral analgesic effects. These models provide a comprehensive assessment of the compounds’ pharmacological profiles, enabling the identification of promising candidates for further development

## 2. Results

### 2.1. Evaluation of Anti-Inflammatory Activity

The effects of the test substances on carrageenan-induced paw edema in rats were compared to the control and the reference drug diclofenac sodium. The significant results are presented in Figure 1 and Figure 2.

Diclofenac sodium exhibited consistently low percentages of edema across all time points with highly significant reductions (*p* < 0.001) compared to the control. This consistent anti-inflammatory effect validates the sensitivity of the assay. Test substance **1** demonstrated dose-dependent activity. At 10 mg/kg b.w., no significant reduction in edema was observed. However, at 20 mg/kg b.w., significant reductions were observed at the 2nd (*p* = 0.035) and 3rd hours (*p* = 0.022), though the effect diminished by the 4th hour. The highest dose (40 mg/kg b.w.) further improved anti-inflammatory activity, achieving significant reductions at the 2nd (*p* = 0.008) and 3rd hours (*p* = 0.046), but this effect also decreased at the 4th hour. Substance **1A** exhibited substantial anti-inflammatory activity at 20 mg/kg b.w., with significant reductions in edema at the 2nd (*p* = 0.005), 3rd (*p* < 0.001), and 4th hours (*p* = 0.004). At the 40 mg/kg dose, edema reduction was significant only at the 4th hour (*p* = 0.030), while the 10 mg/kg dose showed no significant effects. Substance **1B** showed no significant anti-inflammatory activity at any dose or time point. Edema percentages remained comparable to the control group, with *p*-values consistently >0.05. Substance **1C** showed limited activity, with no significant reductions in edema at most doses and time points. Notably, at 20 mg/kg b.w., there was an increase in edema at later time points, with percentages reaching 98.24 ± 20.99% by the 4th hour (*p* = 0.992). Substance **1D** did not significantly reduce paw edema at any dose or time point. Edema percentages across all evaluations remained similar to those of the control group, indicating minimal or no anti-inflammatory activity.

### 2.2. Analgesic Activity of Test Substances

The tail flick and paw withdrawal tests were employed to evaluate the analgesic potential of the test substances, using controlled thermal stimulation. The results are summarized in Table 1.

#### 2.2.1. Tail Flick Test

The tail flick test measures the response latency to a noxious thermal stimulus, primarily reflecting centrally mediated analgesic effects. Metamizole, a reference analgesic, significantly increased tail withdrawal latencies at all time points (*p* < 0.001), demonstrating robust analgesic activity. Among the test substances, compound **1A** at 10 mg/kg exhibited a statistically significant increase in latency only at the third hour (*p* = 0.023), indicating delayed analgesic activity.

#### 2.2.2. Paw Withdrawal Test

The paw withdrawal test assesses the response to thermal stimulation applied to the hind paw, providing insights into both peripheral and central analgesic mechanisms. Metamizole significantly prolonged paw withdrawal latency at all the tested intervals (*p* < 0.001), confirming its strong analgesic effect. Substance **1C** at doses of 10 mg/kg and 20 mg/kg displayed significant analgesic activity at the first hour (*p* = 0.04 and *p* = 0.016, respectively), but the effect was not sustained at later time points. This transient activity suggests an initial response that diminishes as the substance is metabolized or eliminated. Conversely, substance **1** at 20 mg/kg demonstrated a significant analgesic effect only at the third hour (*p* = 0.021), indicating a delayed onset of action.

#### 2.2.3. Formalin Test

The formalin test was employed to evaluate the analgesic potential of the test substances, using chemical stimulation. The significant results are presented in Figure 3.

Metamizole, a known analgesic, significantly reduced the paw-licking time during both the first (*p* < 0.001) and second (*p* = 0.002) phases of the formalin test compared to that in the control group. This confirms its well-documented analgesic efficacy and establishes it as an appropriate positive control in this assay. Compound **1** at 10 mg/kg b.w. showed no significant change in paw-licking behavior, indicating a lack of analgesic effect at this dose. At 20 mg/kg b.w., no significant reduction in paw-licking time was observed in the first phase (*p* = 0.158), while a notable increase was observed in the second phase (*p* = 0.932). This suggests that this dose may not effectively modulate the pain response. However, at 40 mg/kg b.w., a significant reduction in paw-licking time (*p* = 0.038) was observed, indicating potential analgesic effects, particularly during the later phase of pain induced by formalin. Compound **1A**, at all doses tested, failed to produce significant reductions in paw-licking time in both phases, indicating no substantial analgesic effects. Compound **1B**, at 40 mg/kg b.w., exhibited a significant reduction in paw-licking time during the first phase (*p* = 0.001), implying a transient analgesic effect, but no effects were noted in the second phase. Compound **1C**, at 20 mg/kg b.w., showed a significant reduction in paw-licking time in the first phase (*p* = 0.026), indicating some early analgesic activity, though no effects were observed in the second phase. Lastly, compound **1D**, at 40 mg/kg b.w., significantly reduced paw-licking time in the first phase (*p* = 0.043) but had no effect in the second phase, suggesting limited analgesic activity early in the inflammatory response.

## 3. Discussion

In the present study, test substance **1** demonstrated dose-dependent anti-inflammatory activity, with notable effects at higher doses, although the response diminished over time. Substance **1A** showed moderate efficacy, while substances **1B**, **1C**, and **1D** exhibited minimal or no effects. In terms of analgesia, compound **1A** showed delayed effects, and compound **1C** had transient analgesia. In the formalin test, compound **1** reduced pain in the later phase, while other compounds showed limited activity. This discussion explores the pharmacological effects of these hydrazone derivatives, considering their structural influences, mechanisms, and test limitations.

### 3.1. Anti-Inflammatory Activity

Substances **1** and **1A** demonstrated significant anti-inflammatory activity in the carrageenan-induced paw edema model, a widely accepted experimental model that distinguishes early and late phases of acute inflammation. The efficacy of these compounds during the 2nd and 3rd hours corresponded to critical points in the inflammatory cascade. Substance **1** exhibited strong dose-dependent activity at both 20 and 40 mg/kg, but its efficacy did not match the sustained effects of the reference drug diclofenac, a benchmark NSAID. Substance **1A**, on the other hand, showed pronounced activity during the 4th hour, suggesting specific action on late-phase mediators such as prostaglandins. Its limited impact during earlier time points indicates a narrower spectrum of activity compared to diclofenac.

The mechanistic insights drawn from the carrageenan model suggest that substances **1** and **1A** likely target prostaglandin-mediated pathways active during the later phases of inflammation. This hypothesis is supported by their time-dependent activity profiles, particularly the enhanced efficacy of substance **1A** during the 4th hour. Both compounds may act through the inhibition of COX enzymes, which are critical for prostaglandin synthesis, or by modulating prostaglandin receptor signaling. Additionally, the observed early-phase activity for substance **1** suggests an ability to interfere with mediators such as histamine, serotonin, or bradykinin, either by inhibiting their release or antagonizing their receptors [9]. These dual-phase effects indicate a versatile mechanism of action that warrants further exploration.

Mohamed Eissa et al. synthesized anthranilic acid derivatives, which demonstrated significant anti-inflammatory activity without ulcerogenic side effects [10]. Salgin-Gökşen et al. reported hydrazones containing 5-methyl-2-benzoxazolinones with strong analgesic and anti-inflammatory properties [11]. Khan et al. described the anti-inflammatory activity of hydrazone derivatives of quinoxalinone [12], while Rajitha et al. found aryl hydrazone derivatives to exhibit notable anti-inflammatory effects [13]. Bhandari et al. synthesized benzylidene hydrazides with substantial analgesic and anti-inflammatory activity [14]. Additionally, Gökçe et al. reported the activity of 6-substituted-3(2*H*)-pyridazinone-2-acetyl-2-(*p*-substituted benzal) hydrazone derivatives [15], and Moldovan et al. synthesized various hydrazone derivatives with promising in vivo anti-inflammatory effects [16].

Novel diclofenac hydrazone derivatives were synthesized and evaluated for their anti-inflammatory activity using the carrageenan-induced paw edema method. Several compounds demonstrated promising anti-inflammatory effects, with some showing significant activity during the 2nd and 3rd hours post-administration, indicating their potential in modulating acute inflammatory responses [17].

For instance, benzothiazine *N*-acylhydrazones were designed by modifying piroxicam, a known NSAID, and were shown to exhibit superior anti-inflammatory and antinociceptive properties compared to the parent drug [18].

Similarly, 6-substituted-3(2*H*)-pyridazinone-2-acetyl-2(*p*-substituted benzal) hydrazones demonstrated potent anti-inflammatory effects, with certain derivatives like 2a outperforming indomethacin, a standard NSAID. The incorporation of various substituents into the hydrazone moiety, such as chlorophenyl and piperazine groups, influenced the compounds’ pharmacological profiles, suggesting that structural modifications can optimize anti-inflammatory potency [6].

Other hydrazone derivatives, such as *N*-(substituted benzylidene)-2-(*N*-(4*H*-1,2,4-triazole-4-yl) benzamido) acetohydrazide derivatives, were synthesized for both anti-inflammatory and antimicrobial activity. Among these, compounds substituted with halogen and dimethylamino groups exhibited significant anti-inflammatory effects. This demonstrated that the presence of certain substituents on the benzylidene and triazole rings could enhance anti-inflammatory potency while potentially reducing the risk of gastrointestinal side effects commonly associated with NSAIDs [19]. Additionally, pyrazine *N*-acylhydrazones were evaluated for their anti-inflammatory and analgesic activities. These compounds showed promising results in animal models of pain and inflammation [20].

Similarly, isatin derivatives exhibited strong anti-inflammatory, analgesic, and antipyretic effects, with compounds substituted with methyl, chloro, and nitro groups at specific positions on the isatin structure showing superior activity [21].

The incorporation of hydrazone groups into diverse chemical frameworks has led to the discovery of additional classes of anti-inflammatory agents. For example, *N*-phenylpyrazolyl-*N*-glycinyl-hydrazone derivatives showed strong anti-inflammatory activity when compared to standard drugs, while derivatives of substituted-*N*-[(1*E*)-substituted phenylmethylidene] benzohydrazides exhibited good in vitro anti-inflammatory activity due to the presence of functional groups like 4-nitro, 4-methyl, and 2-hydroxy on the phenyl ring [6].

It is valuable to compare the efficacy of the studied compounds with other hydrazone derivatives reported in the literature. A direct comparison, summarized in Table 2, highlights the advantages and limitations of substances **1**, **1A**, **1B**, **1C**, and **1D** in relation to other hydrazone derivatives tested for anti-inflammatory effects. This comparison not only provides context for the performance of our compounds but also emphasizes areas for potential improvement.

However, the carrageenan model has limitations that should be acknowledged. While it is a robust tool for studying acute inflammation, it primarily reflects the roles of histamine, serotonin, bradykinin, and prostaglandins, focusing on vascular and exudative responses. It does not fully capture the complexities of chronic inflammation, which involves additional cellular components like lymphocytes, macrophages, and cytokine networks. Additionally, variability in carrageenan dosing, injection site, and environmental conditions can influence the outcomes, potentially affecting reproducibility. The model also does not account for potential systemic effects or toxicity of the tested compounds, limiting its extrapolation to clinical settings [22,23,24]. Future studies should incorporate models such as the complete Freund’s adjuvant (CFA)-induced arthritis model or the collagen-induced arthritis model to evaluate the long-term anti-inflammatory potential of these hydrazone derivatives. Additionally, investigating their effects on pro-inflammatory cytokines like TNF-α, IL-1β, and IL-6, as well as oxidative stress markers, could provide deeper insights into their mechanisms of action in chronic inflammation. Further pharmacokinetic studies will also be necessary to assess their long-term stability and potential for sustained therapeutic effects.

In contrast, substances **1B**, **1C**, and **1D** showed negligible or no significant anti-inflammatory effects across all tested time points. This lack of efficacy could have resulted from multiple factors, including insufficient interaction with key inflammatory mediators, poor bioavailability, metabolic instability, or structural features that hindered effective binding to target proteins such as COX enzymes or prostaglandin receptors. For instance, structural modifications that increase steric hindrance or alter electronic properties may have negatively impacted the compounds’ ability to engage with active sites of these targets. The dose-dependent effects observed with substances **1** and **1A** further underscore the importance of identifying optimal dosing regimens to maximize therapeutic efficacy.

### 3.2. Analgesic Activity

The formalin test and thermal pain assays provided complementary insights into the analgesic properties of these compounds. The formalin test, which distinguishes neurogenic pain in the early phase from inflammatory pain in the late phase, revealed that substances **1** and **1A** were more effective during the late phase. This aligns with their anti-inflammatory profiles and suggests a primary efficacy against pain mechanisms associated with inflammation. Their limited activity during the early neurogenic phase indicates that these compounds do not significantly target nociceptor activation or sodium-channel-mediated pathways [25].

Despite its utility, the formalin test has limitations. It involves a single administration of an irritant, which may not fully mimic clinical pain conditions characterized by sustained or repetitive stimuli. Moreover, the test cannot isolate the contributions of central versus peripheral pain mechanisms. The reliance on behavioral scoring introduces observer bias and potential variability, emphasizing the need for automated or blinded methods [26,27]. These limitations suggest that additional studies employing chronic pain models, such as the CFA-induced arthritis model or neuropathic pain paradigms, could provide more comprehensive insights into the long-term efficacy of these compounds.

Five hydrazone derivatives (H1–H5) were tested in mice models, revealing their antinociceptive effects primarily in the second phase of formalin-induced nociception [28]. In a study by Nocheva et al., a series of N-pyrrolyl hydrazide-hydrazones were synthesized and evaluated for their analgesic activity using the paw pressure and hot plate tests. The compound designated as DI-5g, which contains an isatin carbonyl fragment, demonstrated the most significant analgesic effect, surpassing that of the reference drug metamizole at the 30 min mark [29]. This suggests that specific structural modifications, such as the incorporation of an isatin moiety, can enhance analgesic potency.

The tail flick and paw withdrawal tests, which evaluate central and mixed pain mechanisms, respectively, showed modest and inconsistent effects for these compounds. Substance **1** exhibited delayed but measurable efficacy, whereas substance **1C** demonstrated transient activity, suggesting variability in pharmacokinetic profiles or receptor interactions. The modest effects observed in these thermal pain models suggest that the tested hydrazone derivatives are less effective in modulating central pain pathways, such as those mediated by opioid receptors or descending inhibitory mechanisms [30,31].

The limitations of thermal pain models should also be considered. Both the tail flick and paw withdrawal tests primarily assess acute pain responses, which may not reflect the mechanisms underlying chronic pain states. These tests are sensitive to variations in stimulus intensity, skin thickness, and animal handling, introducing potential confounding factors. Additionally, these models do not distinguish between peripheral and central contributions to the observed analgesic effects, limiting mechanistic insights [32,33]. To address these gaps, future studies should incorporate models of chronic and neuropathic pain, such as the spared nerve injury (SNI) or sciatic nerve ligation models, which better simulate clinical pain conditions. Additionally, electrophysiological recordings or functional imaging techniques could provide a more direct assessment of neural activity associated with analgesic effects. By expanding the scope of investigations beyond acute pain models, future research can clarify the therapeutic potential of these compounds and determine whether their efficacy extends to more complex and clinically relevant pain states.

### 3.3. Structural Considerations

The structural characteristics of the hydrazone derivatives play a crucial role in shaping their pharmacological profiles. Hydrazones are known for their ability to interact with biological targets through hydrogen bonding, π–π stacking, and hydrophobic interactions. These interactions are central to modulating the activity of enzymes, receptors, and other proteins involved in inflammation and pain.

Substance **1** owes its robust anti-inflammatory activity to specific structural features, such as a 4-chlorophenyl substituent at the 5-position of the pyrrole ring, which enhances hydrophobic interactions with COX enzymes. The hydrazone functional group (-CO-NH-N=CH-) provides flexibility and adaptability, allowing effective binding to diverse inflammatory mediators [34]. The presence of an ethyl ester group at the 3-position further enhances lipophilicity, improving membrane permeability and target accessibility. The phenylpropionyl moiety further contributes to stabilization within enzyme binding pockets, enhancing efficacy in both early and late phases of inflammation [35].

Substance **1A** shares several structural attributes with substance **1** but incorporates additional modifications, such as dual ester groups at the 3- and 4-positions of the pyrrole ring. These groups improve lipophilicity and metabolic stability, potentially extending the compound’s duration of action. The extended hydrazineylidene linkage introduces additional hydrogen-bonding capabilities and spatial flexibility, enabling selective interactions with prostaglandin pathways during late-phase inflammation. However, the increased steric bulk of substance **1A** may limit its activity against early-phase mediators, such as histamine and serotonin, contributing to its delayed onset of action [28].

Conversely, substances **1B**, **1C**, and **1D** lack critical structural features required for effective interaction with inflammatory targets.

The presence of an ethoxycarbonyl group and additional methyl substituents at the 3- and 5-positions of the pyrrole ring in substance **1B** likely alters electronic distribution and steric interactions, reducing affinity for inflammatory mediators. The methylene linkage in the hydrazone moiety introduces rigidity, potentially limiting conformational adaptability required for effective target binding. These modifications might also negatively impact pharmacokinetics, reducing systemic availability or target accessibility [36,37].

Substance **1C** incorporates additional methyl groups on the pyrrole ring and a repositioned ethoxycarbonyl group, which may disrupt its interaction with inflammatory mediators. The increased steric hindrance likely reduces binding affinity to COX enzymes or prostaglandin receptors. Moreover, the altered hydrazone moiety with the hydrazineylidene linkage could restrict flexibility, impairing the ability to adapt to diverse active sites. This structural configuration might also reduce membrane permeability, further limiting efficacy [38].

Substance **1D** features extensive methylation and an ethoxycarbonyl substituent, which may alter its interaction profile with key inflammatory mediators. The substitution at the 3-position of the hydrazone moiety, combined with the pyrrole modifications, likely impacts electronic properties and reduces flexibility, leading to weaker or less specific binding to inflammation-related enzymes [39]. These changes could account for the lack of significant activity in both the anti-inflammatory and analgesic models.

The development of hydrazone derivatives with specific functional groups, such as nitro, halogens, and methyl or chloro substitutions, at various positions on the aromatic ring has proven effective in enhancing anti-inflammatory potency while minimizing undesirable side effects. For example, nicotinic acid hydrazides substituted with nitro groups demonstrated promising anti-inflammatory activities, as did benzophenone semicarbazone and acetophenone semicarbazone derivatives, which exhibited moderate anti-inflammatory effects in carrageenan-induced paw edema models [2,6,40].

Hydrazones are being investigated as dual inhibitors of cyclooxygenase (COX) and 5-lipoxygenase, offering potential advantages over traditional NSAIDs. Evidence suggests that the hydrazone moiety acts as a pharmacophore for COX inhibition. Substituents significantly affect activity, with 4-tolyl or 4-fluorophenyl groups being more effective than 4-bromophenyl or 4-*N*,*N*-dimethylaminophenyl. However, replacing the carboxylic acid group of mefenamic acid with an N-arylhydrazone moiety does not enhance anti-inflammatory properties [41].

Compounds with specific structural features have shown notable analgesic and anti-inflammatory effects. For example, derivatives with methoxy groups at the para-position demonstrate superior edema inhibition. Additionally, hydrazone derivatives synthesized from safrole and furoxanyl-*N*-acylhydrazones have exhibited potent analgesic and anti-inflammatory activities, often surpassing standard drugs like indomethacin [41].

The following diagram (Figure 4) was included to illustrate the signal transduction pathways involving the molecules, along with those related to the inflammatory process and its five signs, highlighting the specific levels at which the newly synthesized compounds exert their effects according to the study’s results.

### 3.4. Future Directions

The combination of the carrageenan model and analgesic tests provides a comprehensive overview of the therapeutic potential of these hydrazone derivatives. While substances **1** and **1A** show promise as dual-phase anti-inflammatory agents, their limited efficacy in broader pain models highlights the need for further optimization. Structural modifications to improve membrane permeability, metabolic stability, and interactions with neurogenic pain pathways could enhance their pharmacological profiles. Computational docking studies and structure-activity relationship analyses will be invaluable in identifying key structural features that govern efficacy. Additionally, complementary studies targeting chronic inflammation, alternative pain models, and specific molecular pathways will further elucidate their therapeutic potential.

In conclusion, the observed anti-inflammatory and analgesic effects of the tested hydrazide and hydrazone derivatives are closely linked to their structural characteristics. Substances **1** and **1A** demonstrate the potential of hydrazones to serve as effective agents against acute inflammation, while the inactivity of **1B**, **1C**, and **1D** underscores the importance of structural refinement. Targeted modifications and further pharmacokinetic analyses will be critical in advancing these compounds toward clinical application.

## 4. Materials and Methods

All experiments were approved by the Animal Health and Welfare Directorate of the Bulgarian Food Safety Agency, based on the position of the Ethics Committee, Bulgarian Food Safety Agency, Protocol No. 419/20 December 2024.

### 4.1. Test Substances

Reagents used in the experiments were NaCl 0.9% (Sopharma AD, Sofia, Bulgaria), metamizole sodium amp. 500 mg/mL 2 mL (Sopharma AD, Sofia, Bulgaria), diclofenac sodium amp. 75 mg/3 mL (Hexal AG, Holzkirchen, Germany), lambda-carrageenan (Merck, Darmstadt, Germany), and formalin 0.2% (Merck, Darmstadt, Germany). Metamizole (200 mg/kg b.w.) was utilized as the positive control in analgesic models, while Diclofenac (25 mg/kg b.w.) served as the positive control for the inflammation models. The novel hydrazone compounds were dissolved in saline to prepare 1% solutions. The volume administered was 0.1, 0.2, and 0.4 mL per kg for the respective doses (10, 20, 40 mg/kg b.w.). All substances were administered intraperitoneally to the animals.

#### 4.1.1. Hydrazide/Hydrazone Compounds

The substances used were as follows: 

Ethyl-5-(4-chlorophenyl)-1-(1-hydrazinyl-1-oxo-3-phenylpropan-2-yl)-2-methyl-1*H*-pyrrole-3-carboxylate (**1**);

Diethyl(*E*)-5-((2-(2-(5-(4-chlorophenyl)-3-(ethoxycarbonyl)-2-methyl-1*H*-pyrrol-1-yl)-3-phenylpropanoyl)hydrazineylidene)methyl)-3-methyl-1*H*-pyrrole-2,4-dicarboxylate (**1A**);

Ethyl(*E*)-5-(4-chlorophenyl)-1-(1-(2-((4-(ethoxycarbonyl)-3,5-dimethyl-1*H*-pyrrol-2-yl)methylene)hydrazineyl)-1-oxo-3-phenylpropan-2-yl)-2-methyl-1*H*-pyrrole-3-carboxylate (**1B**);

Ethyl(*E*)-4-((2-(2-(5-(4-chlorophenyl)-3-(ethoxycarbonyl)-2-methyl-1*H*-pyrrol-1-yl)-3-phenylpropanoyl)hydrazineylidene)methyl)-3,5-dimethyl-1*H*-pyrrole-2-carboxylate (**1C**);

Ethyl(*E*)-5-(4-chlorophenyl)-1-(1-(2-((4-(ethoxycarbonyl)-2,5-dimethyl-1*H*-pyrrol-3-yl)methylene)hydrazineyl)-1-oxo-3-phenylpropan-2-yl)-2-methyl-1*H*-pyrrole-3-carboxylate (**1D**).

#### 4.1.2. Synthesis of *N*-Pyrrolylcarbohydrazide (**1**) and Pyrrole Hydrazones (**1A**–**1D**)

The hydrazones were synthesized by reacting a pyrrole hydrazide with substituted pyrrole aldehydes. The aldehydes were chosen to incorporate a second pyrrole heterocycle into the hydrazone molecules, facilitating the analysis of structure–activity relationships. The starting carbohydrazide was obtained via the selective hydrazinolysis of *N*-pyrrolylcarboxylic acid ethyl ester. The target compounds were synthesized according to the procedure presented in Figure 5, and their structures were confirmed using IR, ^1^H-NMR, ^13^C-NMR, and mass spectrometry, with purity verified by HPLC and melting point analysis [42]. The IUPAC names of the compounds were generated using ChemDraw Professional Software, Version 16.0.0.82 (68). The doses of 10, 20, and 40 mg/kg for the compounds *N*-pyrrolylcarbohydrazide (**1**) and pyrrole hydrazone derivatives (**1A**–**1D**) were chosen based on a broader series of novel compounds with a pyrrolic structure. These doses were consistent with those used in previous studies involving similar pyrrolic derivatives, allowing for a more direct comparison across various compounds in the series. Maintaining these dose levels also ensured consistency for future quantitative structure–activity relationship (QSAR) analyses, facilitating more reliable comparisons and aiding in the identification of structural features that correlate with pharmacological activity.

### 4.2. Experimental Design

#### 4.2.1. Experimental Animals

All experiments were performed using six-week-old male Wistar rats, each weighing approximately 150–200 g. The animals were randomly allocated into two sets of 17 parallel experimental groups, with each group comprising eight rats. They were housed under standard laboratory conditions, including a temperature-controlled environment, a 12 h light/dark cycle, and ad libitum access to food and water.

#### 4.2.2. Carrageenan-Induced Paw Edema

To establish baseline values, the volume of the right hind paw was measured in all animals prior to treatment. Subsequently, edema was induced by injecting 0.1 mL of a 1% carrageenan solution, prepared in 0.9% sodium chloride, into the right hind paw of each rat. Control animals received an intraperitoneal injection of 0.1 mL of 0.9% sodium chloride solution immediately after carrageenan administration.

The paw volume of each rat was subsequently measured using a plethysmometer (Ugo Basile, Gemonio, Italy) at the second, third, and fourth hours after the carrageenan injection. The percentage of paw edema was calculated using the following formula:Paw edema (%) = ((*V*_*t*_ − *V*_0_)/*V*_0_) ∗ 100 
where,

V_0_ = mean paw volume before treatment;

V_t_ = mean paw volume at the respective hour.

A reduction in paw swelling compared to the control group was used as an indicator of anti-inflammatory activity [9].

#### 4.2.3. Tail Flick Test

An infrared heat source (Ugo Basile, Gemonio, Italy) was positioned approximately 3 cm from the distal end of the animal’s tail, delivering controlled heat. The tail withdrawal time was automatically recorded in seconds. To prevent tissue damage, the radiation intensity was set at 80 mW/cm^2^, with a maximum heating duration of 15 s. Each animal was tested at 1 h, 2 h, and 3 h intervals following treatment with the test substances. Analgesic activity was determined by the extension of tail withdrawal latency compared to saline-treated control rats [30].

#### 4.2.4. Paw Withdrawal Test

The animal was placed in a Plexiglas container where it was allowed to move freely. After acclimatization, an infrared heat source (Ugo Basile, Gemonio, Italy) was positioned directly beneath one of the rat’s hind paws. Paw withdrawal latency was automatically measured in seconds. To avoid significant injury to the paw, the infrared radiation intensity was set to 50 mW/cm^2^, and the maximum heating time was limited to 30 s. A special filter was used to block the visible spectrum of light to prevent unintended stimulation. Each animal was tested at 1 h, 2 h, and 3 h intervals after receiving the test substances. An increase in paw withdrawal latency relative to the control group was used as an indicator of analgesic activity [31].

#### 4.2.5. Formalin Test

One hour after administering the test substances, 200 μL of a 0.2% formalin solution was injected intradermally into the hind paw of each animal. The duration of paw-licking behavior was recorded during two observation periods: the initial 10 min and a 20–30 min interval. A reduction in paw licking time relative to the control group was interpreted as an indication of analgesic efficacy [25].

### 4.3. Statistical Analysis

Statistical analyses were conducted using IBM SPSS 26.0 software. A one-way ANOVA was employed, followed by Tukey or Games–Howell post hoc tests, depending on the results of Levene’s test for homogeneity of variances and Welch and Brown–Forsythe’s robust tests for equality of means. The normality of data distribution was assessed using the Shapiro–Wilk test. The results are expressed as the arithmetic mean ± standard error of the mean (mean ± SEM), with statistical significance defined as a *p*-value ≤ 0.05.

## 5. Conclusions

This study highlights the potential of hydrazone derivatives, particularly substances **1** and **1A**, as anti-inflammatory agents, with substance **1** showing dual-phase activity and **1A** excelling in late-phase inflammation. Structural features, such as the 4-chlorophenyl group, were critical for efficacy, while steric and electronic variations in **1B**, **1C**, and **1D** likely reduced activity. Limited analgesic effects suggest their specificity for inflammatory pain, with room for optimization to broaden therapeutic applications. Test model limitations, including a lack of chronic and systemic assessments, warrant further studies to refine structures and pharmacokinetics for clinical use.

## Figures and Tables

**Figure 1 molecules-30-01472-f001:**
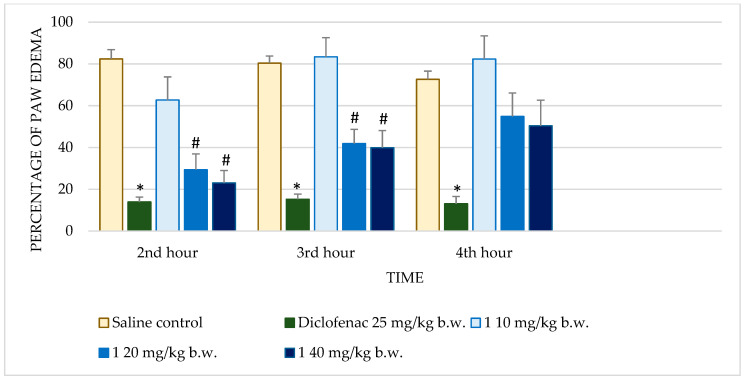
Comparison of edema percentage in carrageenan-induced paw edema model between saline and positive control groups and compound **1**. * *p* < 0.001 compared to control, # *p* < 0.05 compared to control.

**Figure 2 molecules-30-01472-f002:**
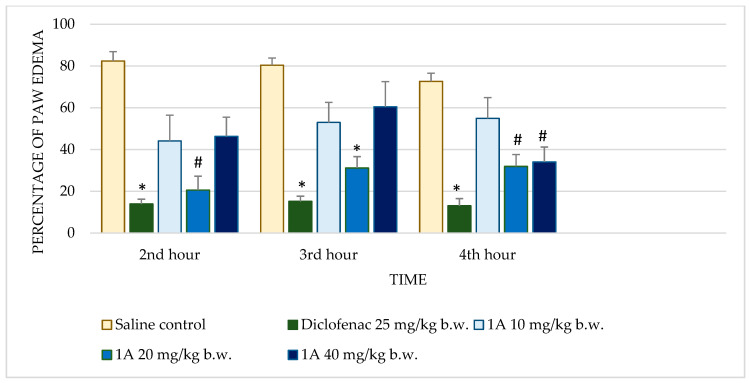
Comparison of edema percentage in carrageenan-induced paw edema model between saline and positive control groups and compound **1A**. * *p* < 0.001 compared to control, # *p* < 0.05 compared to control.

**Figure 3 molecules-30-01472-f003:**
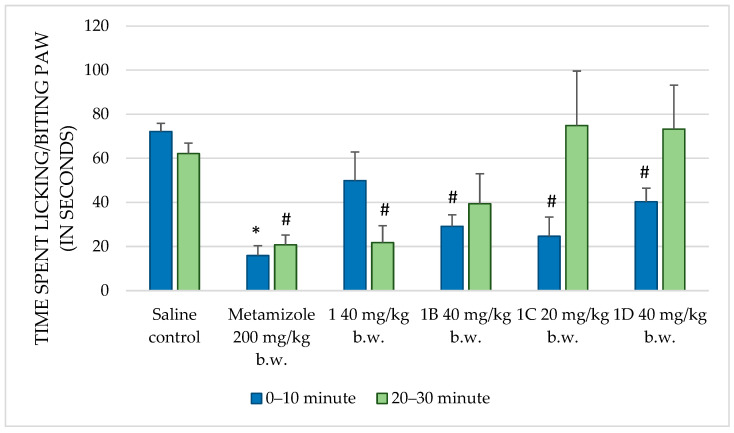
Comparison of the time spent licking/biting paw (in seconds) in formalin test between saline and positive control, and compounds **1**, **1B**, **1C**, and **1D**. * *p* < 0.001 compared to control, # *p* < 0.05 compared to control.

**Figure 4 molecules-30-01472-f004:**
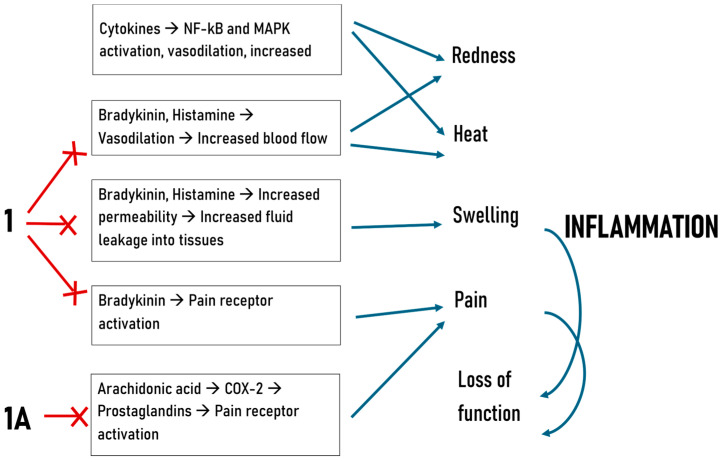
Mechanisms of inflammation and hypothesized action of newly synthesized compounds.

**Figure 5 molecules-30-01472-f005:**
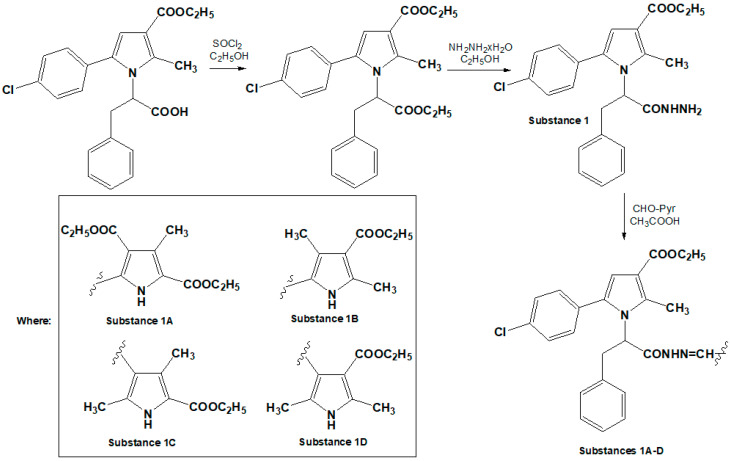
Synthesis pathway of target compounds.

**Table 1 molecules-30-01472-t001:** Comparison of withdrawal latency (in seconds) in tail flick and paw withdrawal tests between control group and groups treated with metamizole.

Experimental Model	Group	Hour	Mean ± SEM	*p*
Tail flick	Control	1st	2.04 ± 0.18	-
		2nd	1.76 ± 0.15	
		3rd	1.66 ± 0.11	
	Metamizole	1st	6.41 ± 0.26	<0.001 *&
		2nd	6.44 ± 0.24	<0.001 *&
		3rd	6.39 ± 0.22	<0.001 *&
Paw withdrawal	Control	1st	7.93 ± 1.01	-
		2nd	9.59 ± 0.99	
		3rd	9.13 ± 0.57	
	Metamizole	1st	17.46 ± 0.84	<0.001 *#
		2nd	19.56 ± 1.10	0.001 *&
		3rd	20.06 ± 1.23	<0.001 *#

Note: * *p* < 0.05 compared to control, # Tukey post hoc, and & Games–Howell post hoc.

**Table 2 molecules-30-01472-t002:** Comparison between compounds **1**, **1A**, **1B**, **1C**, **1D,** and other hydrazone derivatives reported in the literature.

Compound	Anti-Inflammatory Activity	Advantages	Limitations
Substance **1**	Strong, dose-dependent	Effective at higher doses; dual-phase effect	Diminished activity over time; less potent than diclofenac
Substance **1A**	Moderate	Activity in late phase; potential for prolonged action	Limited efficacy in earlier stages
Substance **1B**	Minimal	-	Lack of significant activity
Substance **1C**	Transient	-	Limited and inconsistent effects
Substance **1D**	Negligible	-	Poor bioavailability or receptor interaction
Hydrazone derivatives from literature [10,11,12,13,14,15,16,17,18,19,20,21]	Varies depending on structure	Some exhibit superior anti-inflammatory activity	Structural modifications needed for optimization

## Data Availability

All data generated or analyzed during this study are included in this article.

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
