# Peer review of "Pharmacological Evaluation of Novel Hydrazide and Hydrazone Derivatives: Anti-Inflammatory and Analgesic Potential in Preclinical Models"

_molecules, 2025, doi:10.3390/molecules30071472_

Round 1
Reviewer 1 Report
Comments and Suggestions for Authors
Comments and Suggestions:
1. In the introduction, the authors mention about the compounds “ability to modulate key
biochemical pathways involved in inflammation and pain”, and that “inflammation and pain
are among the most common and debilitating symptoms associated with numerous
pathological conditions, ranging from acute injuries to chronic diseases such as arthritis and
neuropathies. These processes are complex and involve the interplay of pro.inflammatory
cytokines, enzymes, and mediators such as prostlaglandins, histamine, and bradylkinin”
Therefore, I suggest including at the end of the discussion section a comprehensive
diagram that integrates the signal transduction pathways involving the molecules, as well as
those related to the inflammatory process and its five signs.
Additionally, the diagram should indicate at which specific levels the newly synthesized
compounds are acting according to the results obtained in this study. This visual
representation would significantly enhance understanding of the potential mechanisms of
action and provide a clearer context for the observed anti-inflammatory and analgesic
effect.
2. I suggest that tables 1 and 3 be replaced by graphs, to make the observed changes more
evident.
Comments on the Quality of English Language
1. The English Language demonstrates good academic quality; however I suggest reviewing
the structure of the sentences, precision or technical or specific details, cohesion between
paragraphs, concise information and being consistent in the use of terminology.
Editorial Comments
1. The study presents valuable research on the anti-inflammatory and analgesic prpierties of
N-pyrrolylcarbohydrazide and pyrrole hydrazone derivatives, contributing to the
development of new therapeutic agents with improved safety profiles.
2. The methodology employed is robust, using well-stablished models (carrageenan-induced
paw edema, formalin test, tail reflex and paw withdrawal) that provide a comprehensive
assessment of the pharmacological activities.
3. The authors perform a detailed analysis of the structure-activity relationship, clearly
identifying the most promising compounds (1 and 1A) and correlating specific structural
characteristics with the observed effects.
4. The discussion appropriately places the findings in the context of previous research on
hydrazone derivatives, providing a broad perspective on the field.
Recommendations
1. Quantitative data: It is recommended to include in the abstract the edema reduction values
and pain response percentages for the active compounds, which would allow a more direct
comparison with the reference drug (diclofenac).
2. Comparative context: It would be valuable to include a direct comparison of the efficacy of
the studied compounds with other hydrazone derivatives reported in the literature, possibly
through a table highlighting their advantages and limitations.
3. Limitations and perspectives: Although the manuscript mentions limitations of the models
used, it would be advisable to expand the discussion on the implications of these limitations
and suggest additional studies to address aspects not covered in this research, such as
effects on chronic inflammation.
Conclusion
The manuscript presents significant findings on mew pyrrole derivatives with therapeutic antiinflammatory and analgesic potential. With the suggested modifications, especially the inclusion of more specific quantitative data and an integrative mechanistic scheme, the work will constitutive a valuable contribution to the field of medicinal chemistry focused on anti-inflammatory and analgesic agents. I recommend proceeding with the editorial process after making these improvements.
- The English Language demonstrates good academic quality; however I suggest reviewing the structure of the sentences, precision or technical or specific details, cohesion between paragraphs, concise information and being consistent in the use of terminology.
Author Response
"Therefore, I suggest including at the end of the discussion section a comprehensive diagram that integrates the signal transduction pathways involving the molecules, as well as those related to the inflammatory process and its five signs.
Additionally, the diagram should indicate at which specific levels the newly synthesized compounds are acting according to the results obtained in this study. This visual representation would significantly enhance understanding of the potential mechanisms of action and provide a clearer context for the observed anti-inflammatory and analgesic effect."
We have added a comprehensive diagram at the end of the discussion section, as suggested, to integrate the signal transduction pathways involving the molecules, as well as those related to the inflammatory process and its five signs. The diagram also indicates at which specific levels the newly synthesized compounds are acting, based on the results of our study. The updated text and diagram are highlighted in blue.
"I suggest that tables 1 and 3 be replaced by graphs, to make the observed changes more evident."
In response to your suggestion, we have replaced Tables 1 and 3 with graphs to make the observed changes more evident.
"It is recommended to include in the abstract the edema reduction values and pain response percentages for the active compounds, which would allow a more direct comparison with the reference drug (diclofenac)."
We have included the edema reduction values and pain response percentages for the active compounds in the abstract, allowing a more direct comparison with the reference drug (diclofenac), as per your recommendation.
"It would be valuable to include a direct comparison of the efficacy of the studied compounds with other hydrazone derivatives reported in the literature, possibly through a table highlighting their advantages and limitations."
We have included a direct comparison of the efficacy of the studied compounds with other hydrazone derivatives reported in the literature, presented in a new table highlighting their advantages and limitations.
"Although the manuscript mentions limitations of the models used, it would be advisable to expand the discussion on the implications of these limitations and suggest additional studies to address aspects not covered in this research, such as
effects on chronic inflammation."
The discussion on the limitations of the models used has been expanded, and we have added suggestions for future studies to address aspects not covered in this research.
"The English Language demonstrates good academic quality; however I suggest reviewing the structure of the sentences, precision or technical or specific details, cohesion between paragraphs, concise information and being consistent in the use of terminology."
We have consulted with a native speaker and have implemented the suggested changes to enhance the flow and clarity of the manuscript.
Reviewer 2 Report
Comments and Suggestions for Authors
The study has been well designed, organized, worked and prepared. The topic is important and actual.
The points to be improved are detailed below:
- Most of the publications cited in the manuscript are older than 10 years, more attention should be given to the latest research papers review.
-
Authors specify that the double bond of hydrazones derivatives (-CO-NH-N=CH-) is E isomer (line 371). I'd prefer to see the X-ray single analyses some of this structures (hydrazones derivatives 1A-D) to prove this statement.
- In line 371, the name of compound 1A, according to IUPAC rules, certain fragments should be written in italics, for example, as (E) and 1H.
Author Response
"Most of the publications cited in the manuscript are older than 10 years, more attention should be given to the latest research papers review."
The references have been updated and now 30 references (out of 42) are published from 2014 or later.
"Authors specify that the double bond of hydrazones derivatives (-CO-NH-N=CH-) is E isomer (line 371). I'd prefer to see the X-ray single analyses some of this structures (hydrazones derivatives 1A-D) to prove this statement."
The IUPAC names of the compounds were generated using ChemDraw Professional Software, Version 16.0.0.82 (68). We acknowledge that an additional characterization technique, such as X-ray analysis, would help confirm the isomeric form of the synthesized hydrazones. However, due to limited availability of the compounds, this analysis could not be performed.
"In line 371, the name of compound 1A, according to IUPAC rules, certain fragments should be written in italics, for example, as (E) and 1H."
Fragments (E) and 1H are corrected in italics.
Reviewer 3 Report
Comments and Suggestions for Authors
The article Pharmacological Evaluation of Novel Hydrazide and Hydrazone Derivatives: Anti-inflammatory and Analgesic Potential in Preclinical Models, in which I evaluated the anti-inflammatory and analgesic activities of N-pyrrolylcarbohydrazide (1) and four pyrrole hydrazone derivatives (1A–D). The introduction provides an excellent overview of this topic and introduced it effectively. The results section can improve the representation of these.
The discussion is well structured.
I have the following comments.
1. In the materials and methods section, attach an experimental design section.
2. In what were the compounds N-pyrrolylcarbohydrazide (1) and pyrrole hydrazone derivatives (1A–D) dissolved and in what volume were they administered?
3. Why were the doses of 10, 20 and 40 chosen for the compounds N-pyrrolylcarbohydrazide (1) and pyrrole hydrazone derivatives (1A–D)?
4. I suggest that the authors reference in the discussion section the statements they make about the possible mechanism of action of these compounds based on the literature.
5. In my opinion, the results could be more substantial and include graphs and tables.
Author Response
"In the materials and methods section, attach an experimental design section."
Separate tests and experimental models are extensively described and referenced in the manuscript. In response to your suggestion, we have now labeled the section as "Experimental Design" to highlight the details of the study's structure. Could you kindly clarify if you were requesting a graphic or visual representation of the experimental design, or if you were referring to the written description? We would be happy to make any necessary adjustments to meet your expectations.
"In what were the compounds N-pyrrolylcarbohydrazide (1) and pyrrole hydrazone derivatives (1A–D) dissolved and in what volume were they administered?"
The novel hydrazone compounds were dissolved in saline to prepare 1% solutions. The volume administered was 0.1, 0.2, and 0.4 ml per kg for the respective doses (10, 20, 40 mg.kg b.w.). All substances were administered intraperitoneally to the animals. This information is added to the Test substances section of the Materials and Methods and highlighted in blue.
"Why were the doses of 10, 20 and 40 chosen for the compounds N-pyrrolylcarbohydrazide (1) and pyrrole hydrazone derivatives (1A–D)?"
The doses of 10, 20, and 40 mg/kg for the compounds N-pyrrolylcarbohydrazide (1) and pyrrole hydrazone derivatives (1A–D) were chosen based on a broader series of novel compounds with a pyrrolic structure. These doses are consistent with those used in previous studies involving similar pyrrolic derivatives, allowing for a more direct comparison across various compounds in the series. Maintaining these dose levels also ensures consistency for future quantitative structure-activity relationship (QSAR) analyses, facilitating more reliable comparisons and aiding in the identification of structural features that correlate with pharmacological activity. This information is added to the Test substances section of the Materials and Methods and highlighted in blue.
"I suggest that the authors reference in the discussion section the statements they make about the possible mechanism of action of these compounds based on the literature."
We have referenced the statements about the possible mechanism of action of the compounds based on the literature, and the list of references has been updated accordingly.
"In my opinion, the results could be more substantial and include graphs and tables."
We have included graphs, as per the recommendation of other reviewers as well, to further substantiate the results and enhance the presentation of our findings.
Round 2
Reviewer 2 Report
Comments and Suggestions for Authors
The authors have taken into account the comments submitted and have edited the manuscript accordingly, so the manuscript can be accepted for publication.